# Unfiltering of the EarthCARE Broadband Radiometer (BBR) observations: the BM-RAD product

Almudena Velázquez Blázquez[1], Edward Baudrez[1], Nicolas Clerbaux[1], and Carlos Domenech[2]

[1]Royal Meteorological Institute of Belgium, Brussels
[2]GMV, Madrid, Spain

**Correspondence:** Almudena Velázquez Blázquez (almudena.velazquez@meteo.be)

**Abstract.** The methodology to determine the unfiltered solar and thermal radiances from the measured EarthCARE BBR shortwave (SW) and totalwave (TW) filtered radiances is presented. Within the EarthCARE ground processing, the correction for the effect of the BBR spectral responses, the unfiltering, is performed by the so-called BM-RAD processor which produces the level-2 BM-RAD product. The BM-RAD product refers to unfiltered broadband radiances that are derived from the BBR and the Multi-Spectral Imager (MSI) instruments onboard of the forthcoming EarthCARE satellite. The method is based on theoretical regressions between filtered and unfiltered radiances, as is done for the Clouds and the Earth's Radiant Energy System (CERES) and the Geostationary Earth Radiation Budget (GERB) instruments. The regressions are derived from a large geophysical database of spectral radiance curves simulated using radiative transfer models. Based on the radiative transfer computations, the unfiltering error, i.e., the error introduced by the small spectral variations of the BBR instrument response, is expected to remain well below 0.5% in the shortwave (SW) and 0.1% in the longwave (LW), at 1 standard deviation. These excellent performances are permitted by the very simple optics used in the BBR instrument: a telescope with a single paraboloid mirror. End-to-end verification of the unfiltering algorithm has been performed by running the BM-RAD processor on modeled level-1 BBR radiances obtained for three EarthCARE orbits simulated by an integrated forecasting and data assimilation system. The resulting unfiltered radiances are eventually compared to the solar and thermal radiances derived by radiative transfer simulations over the three EarthCARE orbits. In addition, this end-to-end verification has provided further evidence on the high accuracy of the unfiltered radiance process, with accuracies better than 0.5% for SW and better than 0.1% for LW.

## 1 Introduction

The EarthCARE (Earth Clouds, Aerosols and Radiation Explorer) mission (Illingworth et al., 2015; Wehr et al., 2023) is a collaborative mission between the European Space Agency (ESA) and the Japan Aerospace Exploration Agency (JAXA). EarthCARE's primary objective is to enhance our understanding of the processes affecting clouds, aerosols, and radiation in Earth's atmosphere. The mission aims to provide valuable information for improving climate model parametrizations and the understanding of how these components influence the global climate. EarthCARE integrates a suite of instruments including a lidar, radar, and radiometric instruments. Among these instruments, the Broadband Radiometer (BBR) plays the role of

providing crucial information for the radiative closure of the mission. This process involves verifying that the radiative transfer simulations, which are fed with atmospheric products from the mission's active sensors, report radiative fluxes within 10 $\mathrm{Wm}^{-2}$ of the fluxes derived from the BBR.

The BBR will measure accurate shortwave (SW) (0.25 to 4 µm) and totalwave (TW) radiances (0.25 to > 50 µm) at three fixed viewing angles (fore, nadir and aft) along the EarthCARE track. The very fine spatial resolution of the detector array, 648 m along and across-track in nadir, allows to integrate the 3-views over different spatial domains. The radiances measured by the BBR channels are filtered by the spectral response of the instrument, that combines the detector response and the telescope and SW filter throughput. Being directly dependent on the instrument's design, these filtered radiances are of little interest for the science community. They must be converted into (unfiltered) solar and thermal radiances, which are the radiances that would be measured by a perfect instrument, with a flat spectral response, i.e., $\phi(\lambda) = 1$ (where $\lambda$ is the wavelength), that would allow to accurately separate the reflected solar radiation from the Earth's emitted thermal radiation. In the EarthCARE ground processing, this unfiltering process is performed by the BM-RAD processor. In a later stage, the (unfiltered) radiances are converted into hemispheric fluxes, in a second BBR processor called BMA-FLX (Velázquez Blázquez et al., 2024).

The BBR instrument (described in Proulx et al. (2010); Wallace et al. (2009); Helière et al. (2017)) is composed of three telescopes: a fore view at $\pm$ 55° forward, a nadir view at $\pm$ 0°, and an aft view at $\pm$ 55º backward. Any scene located under the satellite track is therefore observed from three directions almost at the same time (about 3 minutes between the fore and aft views). Each telescope uses an array of 30 microbolometer detectors, allowing an across-track swath of $\sim$ 17 km for the nadir view and $\sim$28 km for the two oblique views. The detectors' measurements will be averaged over different spatial domains, namely, *standard*, *small*, and *full*, which are defined by the L1 B-NOM product in the BBR grid (Spilling and Wright, 2020) with an along-track sampling of 1 km. In addition, an additional configurable domain, the *Assessment Domain (AD)*, is defined on the Joint Standard Grid (JSG) for the radiative closure assessment of the EarthCARE mission (Barker et al., 2024). The main inputs to the BM-RAD processor are the level-1 B-NOM product that gives the filtered SW/LW radiances over the standard/small/full domains, the level-1 B-SNG product that gives the filtered SW/TW radiances at detector level, the MSI cloud mask and cloud phase product from M-CLD (Hünerbein et al., 2023), the Joint Standard Grid (X-JSG) and ancillary meteorological data (X-MET) (Eisinger et al., 2024).

The paper is structured as follows. Section 2 presents the spectral response curves of the BBR instrument and introduces the unfiltering problem. Section 3 provides an overview of two large databases of radiative transfer computations that are used to design and parameterize the unfiltering algorithm. Section 4 describes the unfiltering algorithm implemented in the BM-RAD processor. The performances of the algorithm are discussed in Section 5. An end-to-end verification of the algorithm and its implementation in BM-RAD is then presented in Section 6 in which the processor is run on three test scenes of 6200 km each. Final discussion is provided in Section 7.

## 2  BBR spectral responses and unfiltering problem statement

It is not possible to manufacture a broadband radiometer that has perfectly equal sensitivity to the radiation at all wavelengths. The thermal detector elements show some spectral structure in their responses, the throughput of the optics of the instrument also results in spectral variations (Clerbaux, 2008). The signal provided by the instrument, $L_{\text{fil}}$, is a radiance filtered by the spectral response $\phi(\lambda)$ of the instrument:

$$L_{\text{fil}} = \int\limits_0^\infty L(\lambda)\phi(\lambda)d\lambda, \tag{1}$$

where $L(\lambda)$ is the input spectral radiance. For the BBR, the spectral responses of the total and shortwave channels are

$$\phi_{\text{TW}}(\lambda) = \phi_{\text{det}}(\lambda)\,\phi_{\text{tele}}(\lambda), \tag{2}$$

$$\phi_{\text{SW}}(\lambda) = \phi_{\text{det}}(\lambda)\,\phi_{\text{tele}}(\lambda)\,\phi_{\text{quartz}}(\lambda), \tag{3}$$

in which $\phi_{\text{det}}(\lambda)$ is the spectral response of the detectors, $\phi_{\text{tele}}(\lambda)$ is the spectral reflectance of the telescope mirror, and $\phi_{\text{quartz}}(\lambda)$ is the transmittance of the quartz filter used for the SW channel.

Contrary to the SW channel, it is difficult to manufacture an efficient and stable filter to isolate the LW radiation. For this reason, the longwave radiance is obtained by subtracting the SW part in the TW measurement as for the CERES and the GERB instruments. A "synthetic" LW radiance $L_{\text{LW}}$ and spectral response $\phi_{\text{LW}}(\lambda)$ are therefore defined as:

$$L_{\text{LW}} = L_{\text{TW}} - A\,L_{\text{SW}}, \tag{4}$$

$$\phi_{\text{LW}}(\lambda) = \phi_{\text{TW}}(\lambda) - A\,\phi_{\text{SW}}(\lambda), \tag{5}$$

in which the $A$ factor is defined in such a way that the longwave radiance $L_{\text{LW}}$ is equal to exactly zero when an idealized black body solar spectrum of 5800 K is observed:

$$A = \frac{\int_0^\infty L_{5800\text{K}}(\lambda)\phi_{\text{TW}}(\lambda)\,d\lambda}{\int_0^\infty L_{5800\text{K}}(\lambda)\phi_{\text{SW}}(\lambda)\,d\lambda}, \tag{6}$$

where $L_{5800\text{K}}$ is the Planck's emission for a temperature of $T = 5800$ K. As shown in Eq. 6, the factor $A$ is not dependent on the observed scene $L(\lambda)$, but only on the instrument's spectral responses $\phi_{\text{TW}}(\lambda)$ and $\phi_{\text{SW}}(\lambda)$. Figure 1 shows the SW, TW and synthetic LW spectral responses of the BBR instrument for the nadir view of the BBR. The curves for the fore and aft telescopes present only marginal difference with the nadir telescope (not shown).

The filtered radiances ($L_{\text{SW}}$, $L_{\text{LW}}$) are dependent on the instrument characteristics such as the number of mirrors in the optics and type of coating of these mirrors, the type of detector and coating, the thickness of the quartz filter, etc. For this

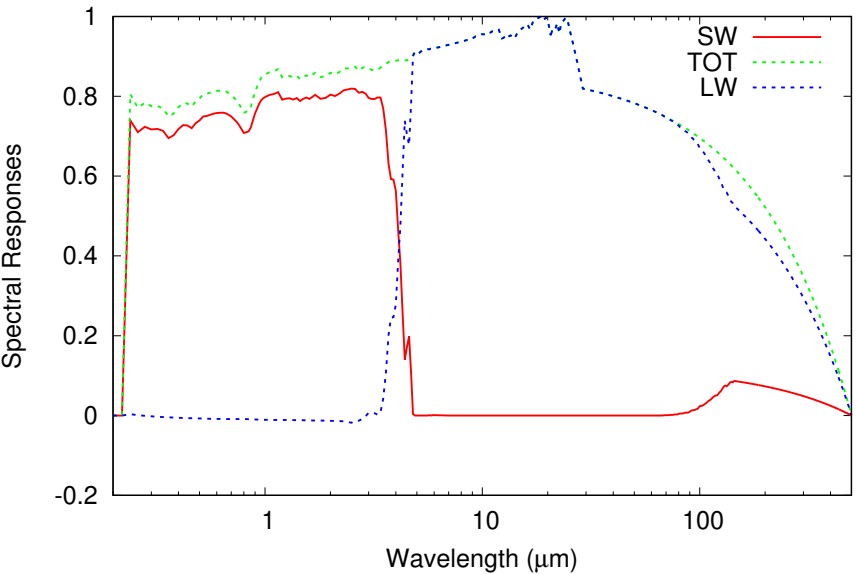

**Figure 1.** EarthCARE BBR spectral responses for the nadir view: shortwave channel $\phi_{\mathrm{SW}}(\lambda)$ (red), total wave channel $\phi_{\mathrm{TW}}(\lambda)$ (green) and synthetic longwave channel $\phi_{\mathrm{LW}}(\lambda)$ (blue).

reason, the filtered measurements have limited scientific interest and they must be converted into unfiltered quantities:

$$L_{\mathrm{sol}} = \int_0^\infty L_{\mathrm{sol}}(\lambda)d\lambda, \tag{7}$$

$$L_{\mathrm{th}} = \int_0^\infty L_{\mathrm{th}}(\lambda)d\lambda, \tag{8}$$

where the distinction is not performed in terms of wavelength $\lambda$ but in terms of type of radiation: $L_{\mathrm{sol}}(\lambda)$ corresponding to
reflection of incoming solar radiation and $L_{\mathrm{th}}(\lambda)$ to thermal emission in the Earth-atmosphere system. This conversion, called *unfiltering*, requires an accurate characterization of the instrument spectral response, $\phi(\lambda)$ and some assumptions about the spectral signature $L(\lambda)$ of the observed scene. Furthermore, it is necessary to estimate the contaminations of the SW channel with thermal radiation and of LW channel with solar radiation. The ratio between the unfiltered and filtered radiances is called either the *unfiltering factor* or the *spectral correction factor*, and are expressed as:

$$90 \quad \alpha_{\mathrm{SW}} = \frac{L_{\mathrm{sol}}}{L_{\mathrm{SW,sol}}} = \frac{\int_0^\infty L_{\mathrm{sol}}(\lambda)\,d\lambda}{\int_0^\infty L_{\mathrm{sol}}(\lambda)\phi_{\mathrm{SW}}(\lambda)\,d\lambda}, \tag{9}$$

$$\alpha_{\mathrm{LW}} = \frac{L_{\mathrm{th}}}{L_{\mathrm{LW,th}}} = \frac{\int_0^\infty L_{\mathrm{th}}(\lambda)\,d\lambda}{\int_0^\infty L_{\mathrm{th}}(\lambda)\phi_{\mathrm{LW}}(\lambda)\,d\lambda}, \tag{10}$$

where SW and LW indicate the BBR spectral channel and sol and th make reference to the kind of radiation, either solar reflected or thermal emitted. Therefore, SW, sol refers to the SW filtered radiances due to solar radiation, while LW, th refers to the LW filtered radiances due to thermal radiation.

In this work, the unfiltering factors are estimated, off-line, from radiative transfer simulations for different scene types on which regressions are derived. These regressions are then used in the BM-RAD processor to infer the unfiltered radiances ($L_{\mathrm{sol}}$, $L_{\mathrm{th}}$) from the filtered measurements ($L_{\mathrm{SW}}$, $L_{\mathrm{LW}}$), where The contamination of both the SW channel by thermal radiation and of the LW channel by reflected solar radiation needs to be estimated prior to the actual unfiltering process. The mathematical forms of these contaminations are:

$$100 \quad L_{\mathrm{SW,th}} \quad = \quad \int_0^\infty \phi_{\mathrm{SW}}(\lambda) L_{\mathrm{th}}(\lambda)\, d\lambda, \tag{11}$$

$$L_{\mathrm{LW,sol}} \quad = \quad \int_0^\infty \phi_{\mathrm{LW}}(\lambda) L_{\mathrm{sol}}(\lambda)\, d\lambda. \tag{12}$$

The thermal contamination in the SW channel, $L_{\mathrm{SW,th}}$, accounts for the planetary thermal emission in the SW channel, below 5 μm and also beyond 50 μm due to the leak in the quartz filters in the far infrared region. The solar contamination in the LW channel, $L_{\mathrm{LW,sol}}$, is generally negative as the synthetic LW spectral response is, very slightly, negative in shorter

wavelengths (below 4 μm). These small quantities should be subtracted from the measured shortwave $L_{\mathrm{SW}}$ and longwave $L_{\mathrm{LW}}$ radiances before the unfiltering process itself can be realized:

$$L_{\mathrm{sol}} \quad = \quad \alpha_{\mathrm{SW}}\, L_{\mathrm{SW,sol}} = \alpha_{\mathrm{SW}}(L_{SW} - L_{\mathrm{SW,th}}), \tag{13}$$

$$L_{\mathrm{th}} \quad = \quad \alpha_{\mathrm{LW}}\, L_{\mathrm{LW,th}} = \alpha_{\mathrm{LW}}(L_{LW} - L_{\mathrm{LW,sol}}). \tag{14}$$

So, the unfiltering process necessitates the estimation of four quantities: the two unfiltering factors ($\alpha_{\mathrm{SW}}$, $\alpha_{\mathrm{LW}}$) and the two

contaminations ($L_{\mathrm{SW,th}}$, $L_{\mathrm{LW,sol}}$).

## 3  Radiative transfer simulations

The unfiltering factors and the contaminations are obtained theoretically from two large geophysical data bases: one of reflected solar radiances containing 5,544 simulations $L_{\mathrm{sol}}(\lambda)$, i.e. 616 unique scenes simulated at 9 Solar Zenith Angles (SZA), and one of Earth's emitted thermal radiances containing 12,096 simulations $L_{\mathrm{th}}(\lambda)$. The solar simulations $L_{\mathrm{sol}}(\lambda)$ are performed for 9

SZA, from 0 to 80 degrees in steps of 10 degrees, and the simulated radiance field is extracted at 18 Viewing Zenith Angles (VZA), $0°$ to $85°$ every $5°$, and 19 Relative Azimuth Angles (RAA), $0°$ to $180°$ every $10°$. These data bases are computed using the LibRadtran 1.4 (Mayer and Kylling, 2005) radiative transfer model as described in Velazquez et al. (2010). The simulations cover a wide range of geophysical conditions and for this purpose, the scene definition has been done using ancillary models and data, such as, surface reflectances from the Aster Spectral Library data (Baldridge et al., 2009) and the Optical Properties

of Aerosols and Clouds (OPAC) software (Hess et al., 1998) for the computation of the aerosol optical properties. The aerosols are assumed to be well mixed and defined in the mixing layer, between 0 and 6 km for desert aerosols and between 0 and 2

km for continental and maritime aerosols. Given that the Aster Spectral Library contains a large number of spectra, a k-means clustering of 12 clusters has been done for clear sky scenes. An averaging of the spectra has been done for those scenes with presence of aerosols. The simulations completely cover the illumination and observation geometries of EarthCARE. Various types of clouds have been simulated with optical thickness from 0.3 to 300, and altitudes ranging from 1 to 12 km. The standard profiles used for the simulations are Tropical, Midlatitude Summer, Midlatitude Winter, Subartic Summer and Subartic Winter (Anderson et al., 1986). Scaling (factor between 0.6 and 1.4) of the water vapor profile in the LW simulations is done to take into account the variability of the water vapor.

Solar simulations have been done in the interval of 0.25 to 5 µm, for 833 wavelenghts, with the following spectral resolution: from 0.25 to 1.36 µm in steps of 0.002 µm, from 1.36 to 2.5 µm in steps of 0.005 µm and from 2.5 to 5 µm in steps of 0.05 µm. Thermal simulations have been done in the interval of 2.5 to 100 µm, for 762 wavelenghts, with the following spectral resolution: from 2.5 to 14 µm in steps of 0.05 µm, from 14.1 to 50 µm in steps of 0.1 µm and from 55 to 100 µm in steps of 0.5 µm. The limit at 100 µm for the simulations is due to the fact that ice and water cloud properties are defined up to this wavelength for both the Yang (Yang et al., 2000) (ice crystals) and Mie (water droplets) parameterizations. As there is still significant radiation beyond 100 µm, the longwave simulations $L_{\text{th}}(\lambda)$ have been extrapolated up to $\lambda = 500$ µm using the black body emission curve corresponding to the brightness temperature simulated at $\lambda = 100$ µm as in Clerbaux et al. (2008b).

The simulated radiances $L(\lambda)$ are convoluted with the SW, TW and LW spectral responses $\phi(\lambda)$ of the BBR for each of the views (fore, nadir, aft) to obtain the filtered radiances for each geometry. The unfiltered radiances are obtained with a perfect constant "filter" $\phi(\lambda) = 1$.

## 4   The BM-RAD algorithm

### 4.1   Flowcharts

In the level-1 B-NOM product, the SW and LW filtered radiances are provided over areas defined according to the instrument grid (the BBR grid). These domains are the *standard*, *small* and *full* integration domains. The level-2 BM-RAD products are then provided at the same domains as the level-1 input. The processing flowchart is given in the left panel of Fig. 2. Firstly, the solar contamination in the LW channel and the thermal contamination in the SW channel are estimated and subtracted from the filtered LW and SW filtered radiances, respectively. In this way purely thermal and solar radiances are obtained. Secondly, the unfiltering factors are estimated and applied to obtain the unfiltered thermal and solar radiances. The level-1 B-SNG product provides measurements of the SW and TW radiation at detector level. The B-SNG file is the main input to compute the level-2 BM-RAD products over the *Assessment Domain* (AD) which is defined on the mission Joint Standard Grid (JSG). Among the different resolutions, the AD is especially important as the EarthCARE radiative computations products (Cole et al., 2023) will be evaluated on this domain. The flowchart is shown in the right panel of Fig. 2, which also shows the additional estimation of the synthetic LW radiance.

Two algorithms have been developed for the shortwave: the "stand-alone" unfiltering that relies only on the BBR observations, and the "MSI-based" unfiltering in which cloud mask and cloud phase information from the MSI are used as additional

information to improve the accuracy of the unfiltering. In the stand-alone algorithm the regression coefficients are dependent on the geometry and surface type while in the MSI-based algorithm they are also dependent on the cloud mask, cloud phase and snow information from X-MET. For the LW only a stand-alone algorithm is implemented, as the unfiltering performs well within the requirements.

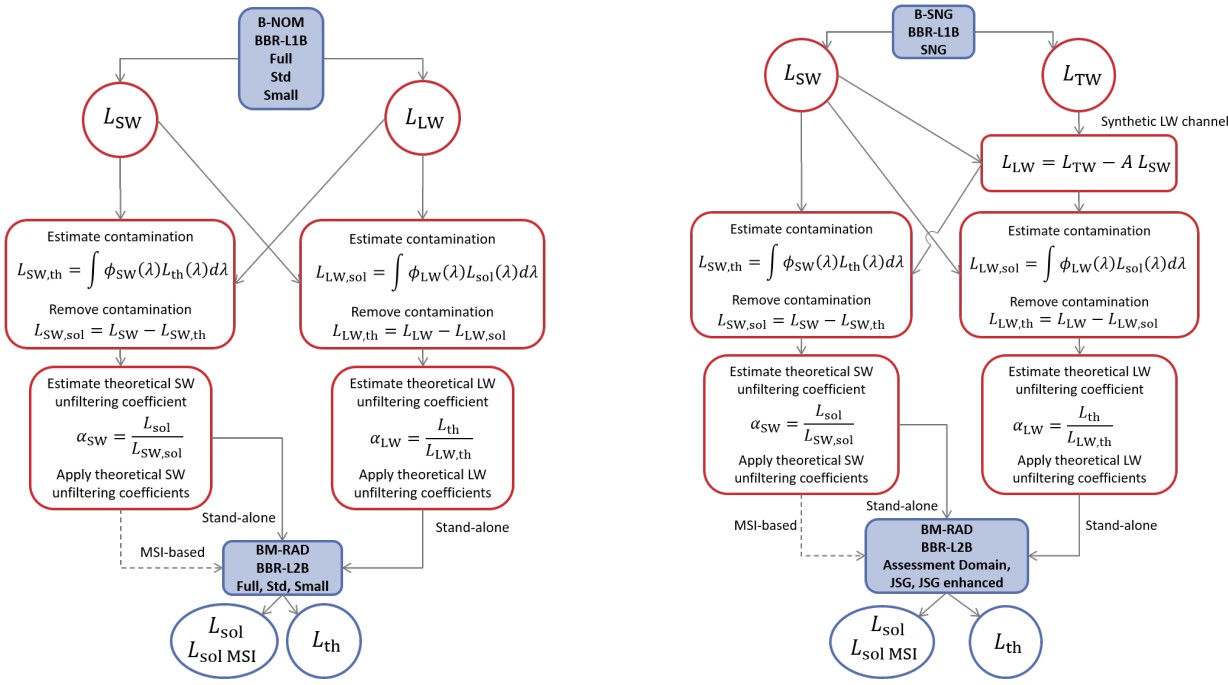

**Figure 2.** Unfiltering flowchart for the BM-RAD product on the BBR grid resolutions (Full, Standard and Small) from the level-1 B-NOM (left panel) and the on the JSG grid resolutions (Assessment, JSG and JSG enhanced) from the level-1 B-SNG product (right panel).

## 4.2  Solar contamination in the LW channel

The Figure 3 shows the scatter plots of the solar contamination in the LW channel, $L_{\mathrm{LW,sol}}$, as a function of the solar radiances, $L_{\mathrm{SW,sol}}$, for the four different surface types (rock, vegetation, ocean, and snow) and for clear and cloudy conditions. Each dot corresponds to one simulation in the libRadtran database. The figures illustrate two particular sun-target-satellite geometries, a sunglint case for the nadir view (SZA=0°, VZA=0°, RAA=0°) and a non- sunglint for the off-nadir views (SZA=30°, VZA=55°, RAA=90°). The contamination of the synthetic LW channel with solar radiation is negative (as expected, as the synthetic LW

response is negative in the shortwave region) and shows a linear relationship with the intensity of the solar radiation. Therefore, the contamination can be estimated as follows:

$$L_{\mathrm{LW,sol}} = a_{\mathrm{cont\_LW}} \, L_{\mathrm{SW,sol}}, \tag{15}$$

where the factor $a_{\text{cont\_LW}}$ is dependent on the geometry (SZA, VZA, RAA) but not on the surface type or the cloudiness. Flat ocean scenes corresponding to sunglint situations have not been considered in the fit (i.e., scenes with sunglint angle lower than 10º and wind speed is equal or lower to 1 m/s) as with the 3 BBR views it will be possible to reduce the sunglint effects as the 3 telescopes will not subject to sunglint at the same time. The residual RMS error of the regression averaged for all the VZA in the database is 0.034 $\text{Wm}^{-2}\text{sr}^{-1}$ which is acceptable for a typical signal in the LW channel ( 60 $\text{Wm}^{-2}\text{sr}^{-1}$). Higher errors are expected for sunglint scenes, in which the error is estimated to be about 0.5 $\text{Wm}^{-2}\text{sr}^{-1}$ for a typical ocean clear sky LW radiance of 80 $\text{Wm}^{-2}\text{sr}^{-1}$.

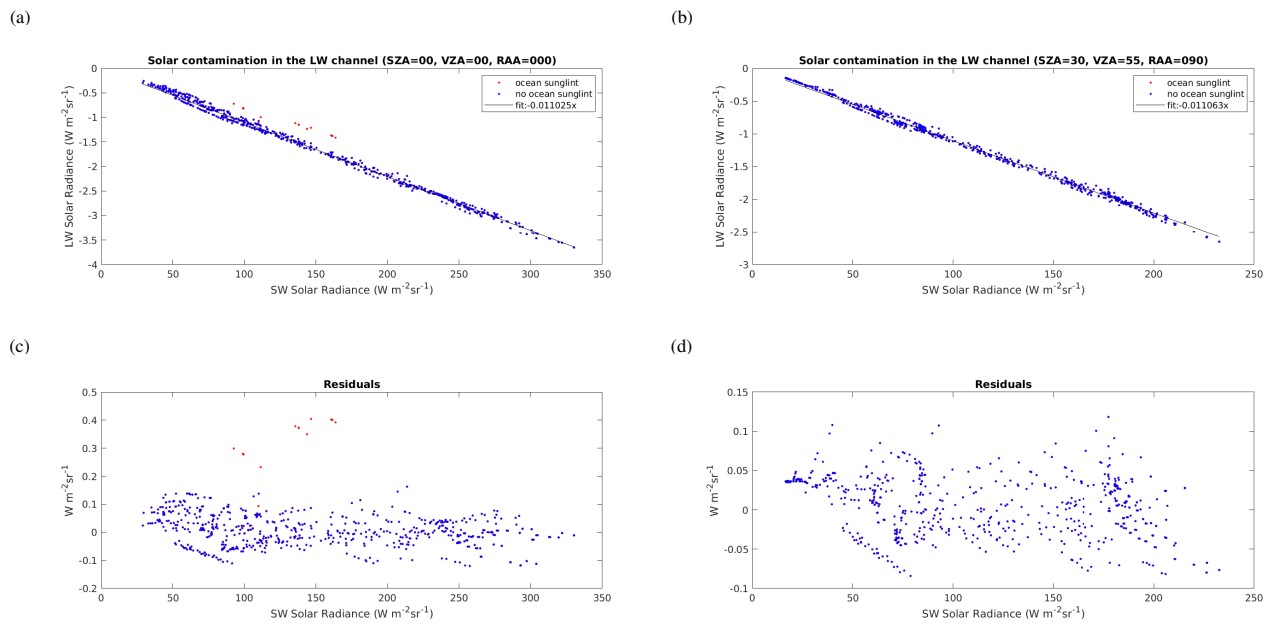

**Figure 3.** Solar contamination in the LW channel. Represented here is the contamination for (a) the nadir view of the BBR for a sunglint geometry (SZA=0°, VZA=0°, RAA=0°), (b) the off-nadir view for a non-sunglint geometry (SZA=30°, VZA=55°, RAA=90°), (c) the residuals of the fit for the nadir view in (a) and (d) residuals for the off-nadir view in (b).

### 4.3 Thermal contamination in the SW channel

The Figure 4 shows the scatter plots of the thermal contamination, $L_{\text{SW,th}}$, as a function of the thermal radiances, $L_{\text{LW,th}}$, for 4 different surface types (rock, vegetation, ocean, and snow) and for clear and cloudy conditions. This figure done for VZA=0° is representative of the nadir view telescope of the BBR. The $L_{\text{SW,th}}$ contamination increases more than linearly with respect to the scene thermal radiance, which is due to the shift of the Planck emission towards shorter wavelengths when the temperature increases. A good fit is obtained with the following relationship:

$$L_{\text{SW,th}} = a_{\text{cont\_SW}} + b_{\text{cont\_SW}} L_{\text{LW,th}}^4, \tag{16}$$

in which the regression coefficients $a_{\mathrm{cont\_SW}}$ and $b_{\mathrm{cont\_SW}}$ are dependent on the VZA, but not on the surface or cloudiness types.

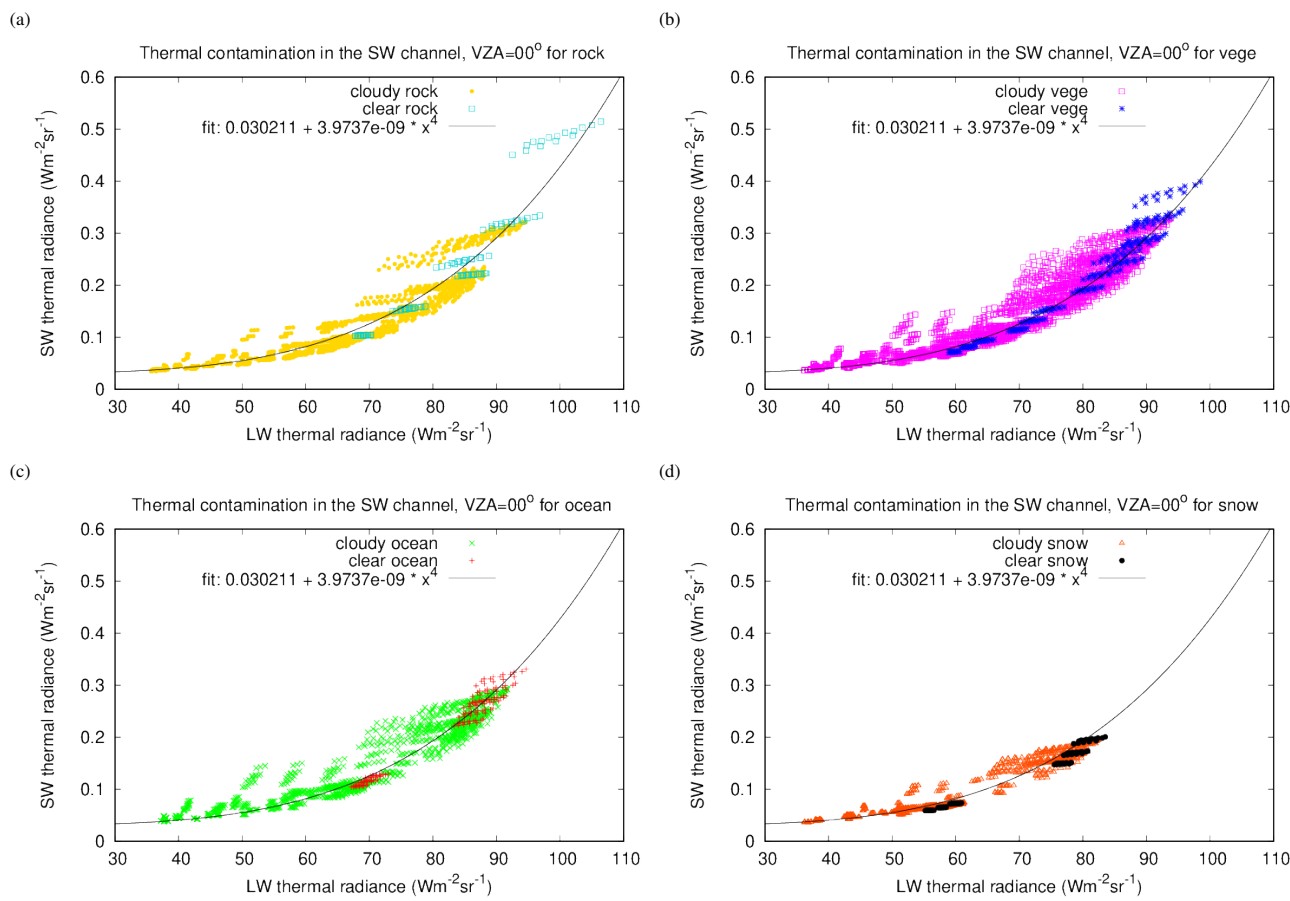

**Figure 4.** Thermal contamination in the SW channel for rock (a), vegetation (b), ocean (c), and snow (d) surfaces. Represented here is the contamination for the nadir view of the BBR. Very similar scatterplots, not shown, exist for the fore and aft views. The polynomial fit shown in the figure is independent of the surface type and cloudiness. For each plot different colour is used to show the clear and cloudy simulations.

For the 12,096 scenes in the LW database, the contamination in the SW channel is lower than 0.6 $\mathrm{Wm}^{-2}\mathrm{sr}^{-1}$. The feather
shape and variability in the LW thermal range comes from the wide range of water content in the water vapour profiles simulated in the LW database. Higher errors in the estimation of the contamination are expected for very warm scenes like bright desert scenes and for scenes with a high water vapour content in warm atmospheres (e.g.,Tropical and Mid Latitude Summer). Ice phase high clouds (at 12 km in the simulations) also show higher errors than the rest of scenes. The residual RMS error on the estimation of the contamination, averaged over all the geometries in the database (VZA from 0º to 85º in steps of 5º), is 0.016
$\mathrm{Wm}^{-2}\mathrm{sr}^{-1}$, which is acceptable with respect to a typical signal in the SW channel of 100 $\mathrm{Wm}^{-2}\mathrm{sr}^{-1}$.

## 4.4 Stand-alone SW unfiltering

A first unfiltering algorithm that does not rely neither on the MSI radiances nor on cloud products has been developed. In the flowchart of Fig. 2 this step corresponds to the $\alpha_{\text{SW}}$ estimation boxes. The motivation behind this is to enable the BBR data to be unfiltered even if the MSI observations are unavailable or if they become degraded with time. In addition, the stand-alone unfiltering algorithm may be useful to assess the problems introduced by the cloud parallax between the fore and aft views and the MSI nadir observation of the scene. An example of the distribution of the unfiltering factors is given in figure 5 (top left panel) for a given geometry (SZA=30º, VZA=55º, RAA=90º) representative of the fore and aft views.

The other panels in figure 5 show the same data but separated according to the surface type (rock, vegetation, ocean, snow and soil). For most of the surface types, the best fit is obtained with the hyperbolic equation:

$$\alpha_{\text{SW}} = a_{\text{SW}} + \frac{b_{\text{SW}}}{L_{\text{SW,sol}}}, \tag{17}$$

which is identical to the $L_{\text{sol}} = b_{\text{SW}} + a_{\text{SW}} L_{\text{SW,sol}}$ relation used for the CERES (Loeb et al., 2001) and GERB (Clerbaux et al., 2008a) shortwave channels. It is worth to mention that the CERES team is currently reviewing its unfiltering process and several improvements are proposed in Liang et al. (2023) for possible inclusion in Edition 5. Those improvements concern mostly the use of the Cox-Munk ocean BRDF, MODIS BRDF over land, seasonal variations of the vegetation, a finer angular resolution and the use of MODTRAN 5.2 for the radiative transfer simulations. Future versions of the BBR unfiltering could potentially benefit from this revision of the original CERES Unfiltering.

The regression coefficients $a_{\text{SW}}$ and $b_{\text{SW}}$ are dependent on the surface type and on the viewing and solar geometries. The residual RMS error of the fit is provided in the different panels of Fig. 5. The typical RMS error of 0.004 $\text{Wm}^{-2}\text{sr}^{-1}$ corresponds to about 0.3% relative error on the unfiltering factor $\alpha_{\text{SW}}$, thus also to 0.3% on the unfiltered radiance $L_{\text{sol}}$, and later also on the solar flux $F_{sol}$.

## 4.5 MSI-based SW unfiltering

The objective of this scene dependent unfiltering is to further reduce the unfiltering error (evaluated at 0.3% at 1 standard deviation for the stand-alone, see previous section) using explicit information about the scene type within the BBR domains (*standard*, *small*, *full* or *Assessment Domain*). The study is similar to the stand-alone one, but the regressions (see equation 17) are in this case also dependent on the cloud mask (clear or cloudy condition) and the cloud phase (water droplets or ice crystals). Although the MSI-based algorithm provides slightly better results in the validation than the stand-alone algorithm (plots not shown in the text), its applicability to the fore and aft views might be affected by cloud parallax effects, as the MSI provides a nadir view of the scene. The Table 2 presents the results obtained for test scenes in comparison with the results derived from the stand-alone approach. The applicability of the MSI-based algorithm will be tested during the commissioning phase once real data will be available.

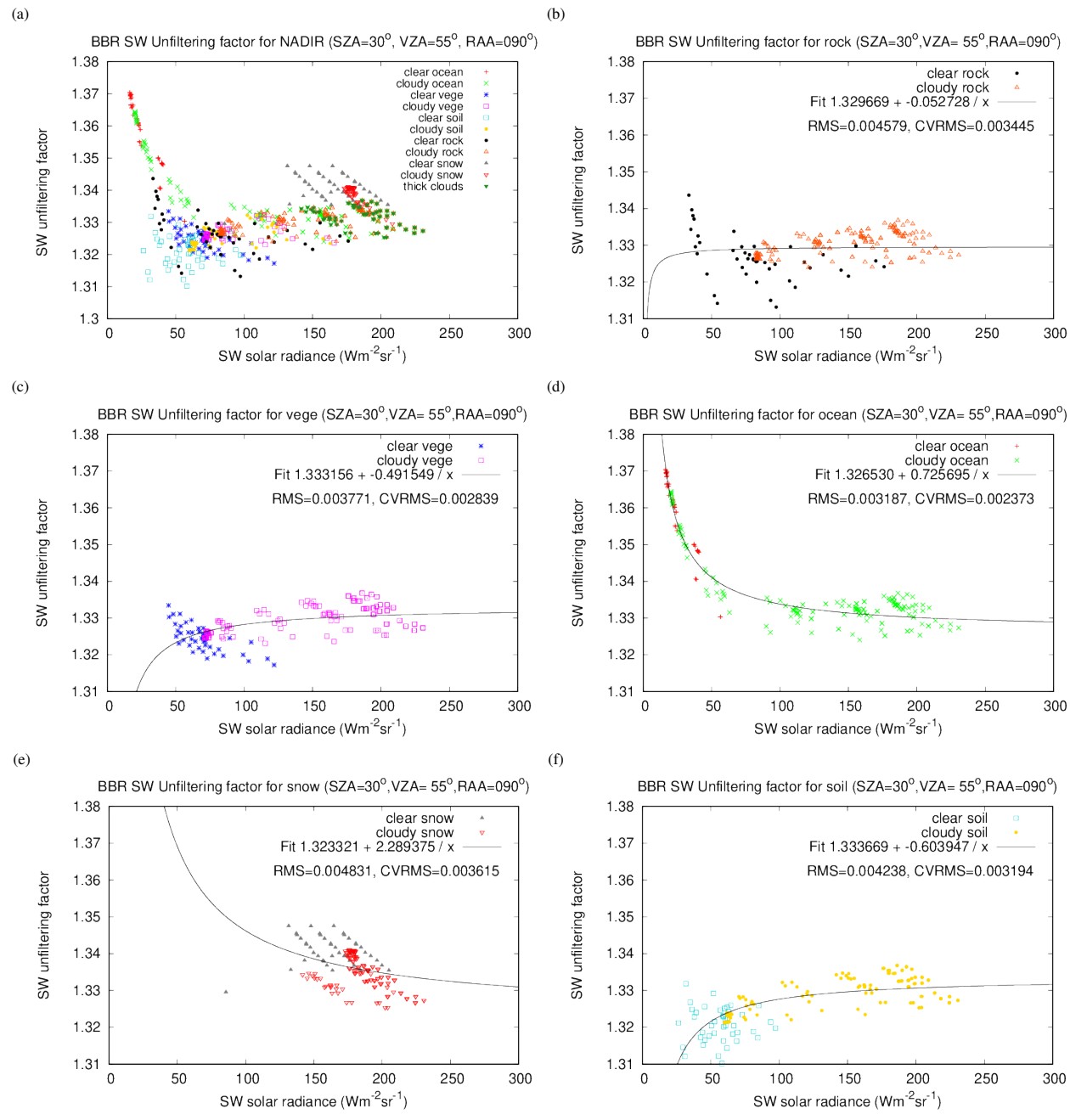

**Figure 5.** Stand-alone SW unfiltering for all scenes together (a) and then separated according to the surface type (rock (b), vegetation (c), ocean (d), snow (e) and soil(f)). These graphs are for the (SZA=30º, VZA=55º, RAA=90º) geometry, representative of the fore and aft views. Similar graphs exist for the nadir view (not shown).

### 4.6  LW unfiltering

The LW channel unfiltering is illustrated in Fig. 6 which shows the scatterplots of the unfiltering factor $\alpha_{lw}$ versus the longwave thermal radiance $L_{th,lw}$, for the nadir view (left panel) and the VZA=55° views (right panel). The range of variability of the LW unfiltering factor for the BBR instrument is very reduced and much smaller than for the CERES and GERB instruments. The primary reason for this is that the BBR optics has only one mirror while CERES has two and GERB has five. A second degree polynomial fit in the scatterplots appears suitable to estimate the unfiltering coefficients:

$$\alpha_{lw} = a_{\text{LW}} + b_{\text{LW}} \, L_{LW,th} + c_{\text{LW}} \, L_{LW,th}^2, \tag{18}$$

where the coefficients $a_{\text{LW}}$, $b_{\text{LW}}$, and $c_{\text{LW}}$ are dependent on the VZA. For clear sky warm scenes, the LW unfiltering factor presents enhanced variability due to variability in the spectral emissivity of the desert surfaces. Even though the performance of a single regression, with a RMS of about 0.1 %, is sufficient with respect to the scientific requirements of the mission, it was investigated if any improvement could be obtained using specific regressions for ocean, vegetation and desert surfaces. The improvement of using surface type dependent regressions is negligible and is therefore not further considered.

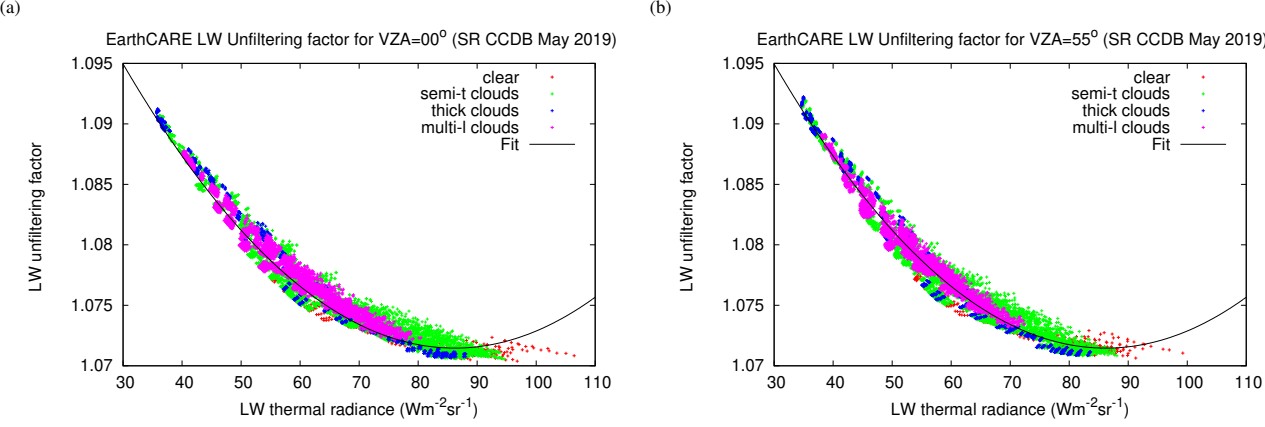

**Figure 6.** Scatter plot of unfiltering factors for the LW channel of the BBR (see Eq. 18), for (a) VZA=0° and (b) VZA=55°

.

### 5  BM-RAD algorithm verification

An analysis of the unfiltering error is performed for 10 typical scene types covering the full extent in terms of SW and LW radiances. The error combines the one due to the estimation of the contamination and the error due to the estimation of the unfiltering factor. The results are summarized in Table 1 which provide values averaged over the full range of simulated SZA, VZA and RAA geometries. For the solar radiation, the relative error on the unfiltered radiances is $\approx 0.26\%$ for cloudy conditions, and increases up to 0.34 % for clear sky conditions. For the thermal radiation, the relative error is $0.10 \pm 0.02$ % for all of scene conditions.

**Table 1.** BBR unfiltering error analysis for 10 scene types (first column). For each scene type, the columns give: the number of libradtran simulations in the solar data base, the averaged solar radiance $<L_{\rm sol}>$ and the RMS error on this radiance due to the subtraction of the thermal contamination and the error due to the unfiltering of the shortwave channel. The RMS error is expressed as an absolute value, in $\rm Wm^{-2}sr^{-1}$, and also as a relative error, in %. The last 4 columns provide the results for the unfiltering of the LW channel. Averaged values have been calculated over all the observation and illumination geometries, i.e., SZA [0:10:80], VZA [0:5:85] and RAA[0:10:180]

| Type of scene | $N°$ of solar scenes | $<L_{\rm sol}>$ $\rm Wm^{-2}sr^{-1}$ | RMS $L_{\rm sol}$ $\rm Wm^{-2}sr^{-1}$ | (%) | $N°$ thermal scenes | $<L_{\rm th}>$ $\rm Wm^{-2}sr^{-1}$ | RMS $L_{\rm th}$ $\rm Wm^{-2}sr^{-1}$ | (%) |
|---|---|---|---|---|---|---|---|---|
| thick high cloud | 36 | 208.84 | 0.53 | 0.26 | 135 | 53.60 | 0.053 | 0.092 |
| semi high cloud | 120 | 117.34 | 0.30 | 0.26 | 702 | 61.07 | 0.058 | 0.088 |
| semi high water cloud | 60 | 117.77 | 0.30 | 0.26 | 702 | 61.07 | 0.059 | 0.090 |
| semi high ice cloud | 60 | 116.9 | 0.30 | 0.26 | 702 | 56.22 | 0.053 | 0.087 |
| semi low cloud | 60 | 112.53 | 0.29 | 0.26 | 702 | 72.19 | 0.063 | 0.082 |
| thick low cloud | 18 | 194.55 | 0.50 | 0.26 | 135 | 70.59 | 0.057 | 0.076 |
| clear desert | 43 | 89.74 | 0.29 | 0.32 | 90 | 78.96 | 0.079 | 0.093 |
| clear ocean | 56 | 39.4 | 0.10 | 0.26 | 135 | 77.21 | 0.075 | 0.091 |
| clear snow | 43 | 172.31 | 0.59 | 0.34 | 90 | 65.53 | 0.078 | 0.112 |
| clear vege | 43 | 75.5 | 0.21 | 0.27 | 225 | 74.85 | 0.079 | 0.099 |

## 6  End-to-end verification of the algorithm using test scenes

In this section, data from the three EarthCARE test frames (Qu et al., 2023; Donovan et al., 2023) have been used to verify the performances of the BM-RAD processor. In general a close agreement is found between the unfiltered radiances calculated by the BM-RAD processor and the reference radiances (truth) obtained directly by broadband integration of the radiative transfer computations on the Global Environmental Multiscale Model (GEM) scenes. Table 2 details the results in terms of bias, RMS and standard deviation, all expressed in $\rm Wm^{-2}sr^{-1}$, and this for the 3 telescopes (fore, nadir, aft) and for the three test frames (Halifax, Baja, Hawaii).

The Table 2 summarises the performances of the stand-alone SW unfiltering, MSI-based SW unfiltering and stand-alone LW unfiltering procedures for the three test scenes. The error metrics show that the MSI-based shortwave unfiltering provides in general a small improvement. The gaining of including MSI information in the unfiltering process while improving results might not be very large in practice because of parallax effects. Another interesting finding is that the unfiltering of the nadir view is, in general, more accurate than the one of the fore and aft views (with, however, an exception for the stand-alone shortwave unfiltering for the Hawaii scene).

The Fig. 7 shows in the upper panels the filtered, unfiltered and simulated (truth) radiances along the orbit frame for the three test scenes. Differences between between filtered and unfiltered radiances can be clearly seen. As expected greatest differences are observed over cloudy scenes in the SW regime, while clear-sky scenes present the higher differences in the thermal radiances. Lower panels show the detail of the differences between the unfiltered radiances and the truth radiance. The corresponding mean difference (bias), standard deviation and RMS error are provided to quantitatively analyse the comparison. The complete summary of results is available in Table 2. The RMS error for both SW and LW unfiltered radiances are well within the accuracy requirement for the BBR that is 2.5 $\rm Wm^{-2}sr^{-1}$ for the SW and 1.5 $\rm Wm^{-2}sr^{-1}$ for the LW. It is worth

**Table 2.** Statistics of the BBR unfiltering errors for the 3 test scenes (Halifax, Baja, Hawaii) and each of the 3 views of the BBR (fore, nadir, aft). The upper parts of the table provide the errors for the stand-alone and MSI-based unfiltering of the SW channel. The bottom part is for the (stand-alone) unfiltering of the longwave channel.

| Scene | Halifax | | | Baja | | | Hawaii | | |
|---|---|---|---|---|---|---|---|---|---|
| stand-alone SW | fore | nadir | aft | fore | nadir | aft | fore | nadir | aft |
| RMSE ($\mathrm{Wm^{-2}sr^{-1}}$) | 0.4864 | 0.3180 | 0.3787 | 1.0365 | 0.7305 | 0.8266 | 0.3729 | 0.4039 | 0.4159 |
| stddev ($\mathrm{Wm^{-2}sr^{-1}}$) | 0.4253 | 0.2223 | 0.2996 | 0.8991 | 0.6773 | 0.7264 | 0.3289 | 0.3006 | 0.3806 |
| bias ($\mathrm{Wm^{-2}sr^{-1}}$) | -0.2361 | -0.2273 | -0.2317 | 0.5158 | 0.2737 | 0.3945 | -0.1757 | -0.2697 | -0.1676 |
| MSI-based SW | fore | nadir | aft | fore | nadir | aft | fore | nadir | aft |
| RMSE ($\mathrm{Wm^{-2}sr^{-1}}$) | 0.4616 | 0.3142 | 0.3817 | 1.0071 | 0.7926 | 0.9197 | 0.3995 | 0.3852 | 0.4030 |
| stddev ($\mathrm{Wm^{-2}sr^{-1}}$) | 0.4014 | 0.2168 | 0.2869 | 0.9421 | 0.7494 | 0.8364 | 0.3652 | 0.066 | 0.3727 |
| bias ($\mathrm{Wm^{-2}sr^{-1}}$) | -0.2279 | -0.2275 | -0.2518 | 0.3557 | 0.258 | 0.3825 | -0.1621 | -0.2332 | -0.1534 |
| LW | fore | nadir | aft | fore | nadir | aft | fore | nadir | aft |
| RMSE ($\mathrm{Wm^{-2}sr^{-1}}$) | 0.1601 | 0.1052 | 0.1169 | 0.2387 | 0.2460 | 0.2210 | 0.3114 | 0.3432 | 0.3160 |
| stddev ($\mathrm{Wm^{-2}sr^{-1}}$) | 0.15223 | 0.0903 | 0.1061 | 0.2368 | 0.2431 | 0.2191 | 0.3015 | 0.3323 | 0.3063 |
| bias ($\mathrm{Wm^{-2}sr^{-1}}$) | 0.0498 | 0.0539 | 0.049 | 0.0306 | 0.0377 | 0.029 | 0.0776 | 0.0861 | 0.0777 |

to mention that these metrics are likely an overestimation of the errors because of the simplifications needed in the radiative transfer computations used for the construction of the test scenes.

## 7 Conclusions

In this paper, the algorithms used by the unfiltering processor (BM-RAD) for the BBR instrument onboard EarthCARE are described. The main output of BM-RAD are the unfiltered solar and thermal radiances for the three BBR views integrated over different spatial domains. These radiances are the main input for the BMA-FLX processor in which the three views are combined to estimate the hemispheric outgoing shortwave and longwave radiative fluxes.

Thanks to its design, the BBR instrument sensitivity shows limited spectral variability which is a pre-requisite for an accurate unfiltering process. The typical stand-alone unfiltering errors are expected to be approximately 0.5% for the shortwave channel and well below 0.1% for the longwave channel. The implementation of the algorithm has been successfully verified on the three EarthCARE test scenes (Halifax, Baja and Hawaii).

Scene information from the MSI radiances (from M-RGR product), MSI cloud retrievals (from M-CLD processor), or snow products (from X-MET product) are useful to further reduce the unfiltering error. So, in addition to the stand-alone unfiltered radiances, the BM-RAD product also contains the MSI-based unfiltered radiance for the shortwave radiation (the improvement for the longwave radiance was considered negligible and therefore not included in the product). However, the MSI-based unfiltering of the fore and aft views might suffer from the parallax effect as the MSI provides only nadir observations. Therefore,

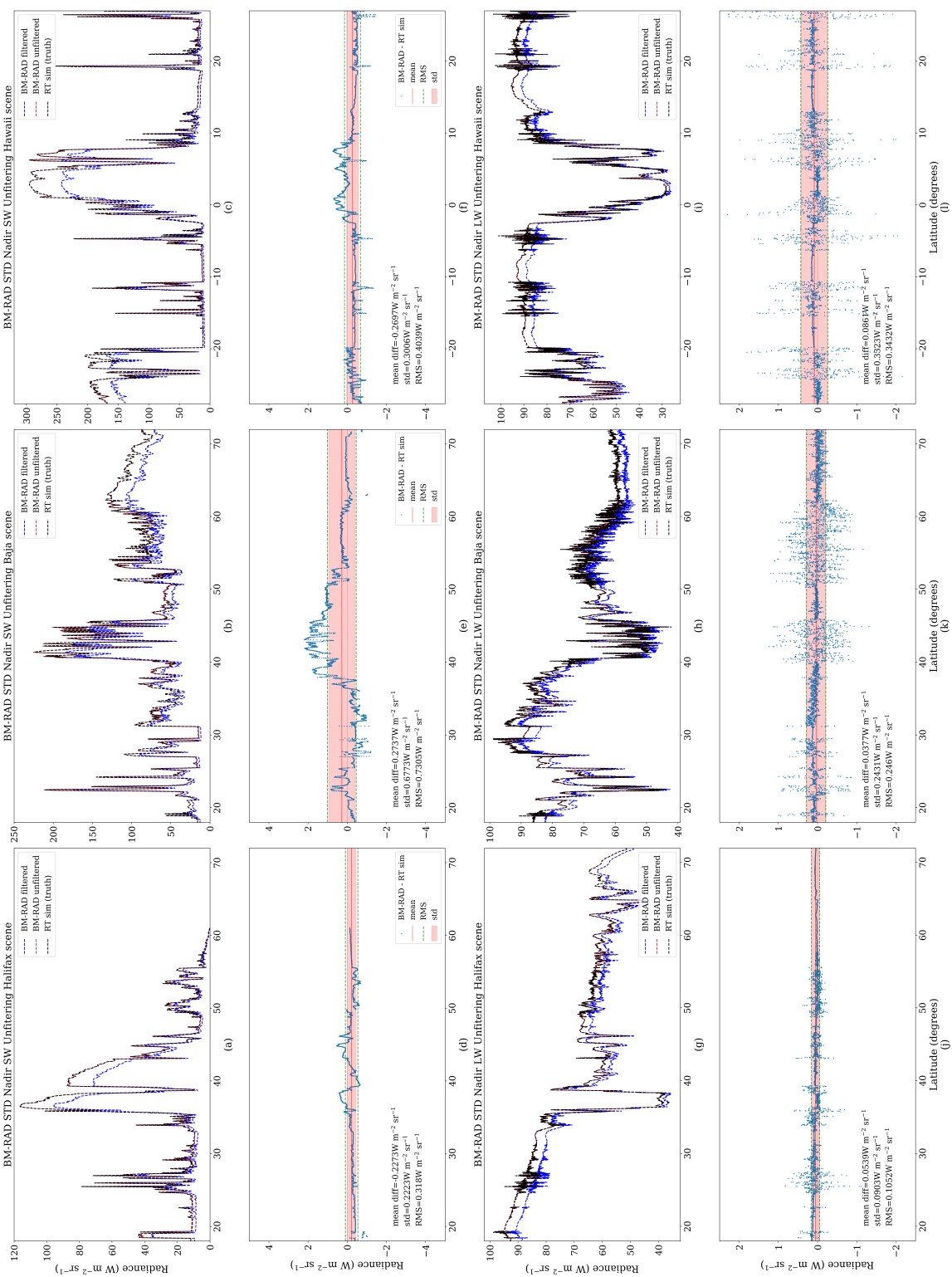

**Figure 7.** Filtered and unfiltered radiances versus the radiative transfer simulations "RT sim (truth)" across the 3 test scenes of Halifax (a,g), Baja (b,h) and Hawaii (c,i). The (a,b,c) panels show the shortwave radiances along the satellite track (and goes to zero in night time condition). The (g,h,i) panels show the corresponding results for the longwave radiation. The difference between the BM-RAD unfiltered solar radiance and the reference value is shown in (d,e,f) and the corresponding differences for the thermal radiances are shown in (j,k,l). The graphs shown here correspond to the nadir view. Similar results are obtained for the off-nadir views (not shown).

a comprehensive evaluation of the MSI-based unfiltering is foreseen to be carried out during the commissioning phase, when the BM-RAD processor can be applied on real BBR and MSI observations. In the meantime, the stand-alone unfiltered radiances should be used.

*Data availability.* The EarthCARE demonstration products from the simulated scenes, including B-NOM and B-SNG L1 data and the BM-RAD L2 products discussed in this paper are available from https://doi.org/10.5281/zenodo.7728948 (van Zadelhoff et al., 2023). The radiative transfer simulations database and description is available at https://gerb.oma.be/public/almudena/SITS_DB_compressed/

*Author contributions.* The manuscript was prepared by AVB, EB, NC and CD. The BM-RAD software was developed by AV, EB and NC

*Competing interests.* The authors declare that they have no conflict of interest

*Acknowledgements.* This research has been funded by the European Space Agency (ESA) (ESA Contract No. 4000112019/14/NL/CT (CLARA) and 4000134661/21/NL/AD (CARDINAL)). We express our deepest gratitude to Dr. Tobias Wehr, who sadly passed away on February 1, 2023, for his continuous support, contagious positivism and endless discussions. EarthCARE would not have been possible without the contributions of Dr. Wehr.

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
