# Peer review of "Unfiltering of the EarthCARE Broadband Radiometer (BBR) observations: the BM-RAD product"

_Atmospheric Measurement Techniques, 2023_

## Referee Comment (RC1)

**Review of "Unfiltering of the EarthCARE Broadband Radiometer (BBR) observations: the BM-RAD product" by Velázquez Blázquez et al.**

**27 September 2023**

**General comments**

The manuscript prepared by Velázquez Blázquez et al. presents an approach to derive unfiltered shortwave (SW) and longwave (LW) radiances from the EarthCARE Broadband Radiometer (BBR). This is an essential processing step to remove unwanted features in the directly measured radiances that are associated with the instrument spectral response. The approach follows that of existing CERES and GERB broadband radiometer measurements that rely on spectral radiance databases simulated by a radiative transfer model. The errors associated with the unfiltering process are reported to be below 0.5% and 0.1% in the SW and LW, respectively.

The paper will serve as a useful reference within the scientific literature for any future users of the EarthCARE BBR data. Overall the paper is clear and logical, well written, and supported with appropriate figures. I noticed some inconsistencies with the EarthCARE overview paper that is also part of this special issue. There are several other aspects that I think would benefit from further clarifications and explanations, especially regarding the radiative transfer simulations. There is also a new paper for CERES unfiltering that should be considered. After addressing these concerns, as outlined in my comments below, I recommend publication in AMT.

**Specific comments**

L20: In Wehr et al. 2023, the spectral limits of the BBR SW and TW channels are stated as < 0.25 to 4.0 μm for SW and 0.25 to > 50 μm for TW. Here, the spectral limits are stated as 0.25 to 5 μm for SW and 0.25 to 500 μm for TW. Since both studies are new, I expect the limits have not changed, rather there is an error somewhere. In most other studies, this might seem like a picky comment because there is probably very little energy difference between the two sets of spectral limits. However, since the purpose of this study is spectral unfiltering, the instrument spectral limits seem like a basic characteristic to ensure is correct and consistent.

L21: Similar to the comment above, the stated spatial resolution of the detector array (700m along track and 600 m across track) is inconsistent with that stated in Wehr et al. 2023 (648 m along track, 800 m across track). Please check that stated numbers are correct and consistent. Do these numbers represent resolution, or sampling distance? It also should be noted that these numbers are relevant to nadir only.

L31: I calculate that, at an EarthCARE altitude of 393 km with an orbital period of 92.5 min, the duration between fore and aft 55° views of the surface is 2.79 minutes. If the authors agree, it would be better to update "about 2 minutes" to "about 3 minutes".

L32: Similar to the first two comments above, the stated swath of the detector array (∼ 17 km for the nadir view and ∼28 km for the two oblique views) is inconsistent with that stated

in Wehr et al. 2023 (±10.2 for nadir and ±16 for off-nadir views). Please ensure values are correct and consistent. This is information that data users are likely to pick up on and, once published, incorrect numbers can be easily propagated into other works.

L28: There is a reference to another paper that describes the BMA-FLX processor (Velázquez-Blázquez et al., 2023), but as far as I can tell this paper is not available anywhere online. It would have been good for the reviewers to at least see a draft copy of this paper if it is to be cited here.

L59-60: For the CERES instrument flying on the NOAA-20 satellite, there is a dedicated LW channel. So, the statement that LW is calculated by subtraction is not *always* true.

Equation 6: What is the value of "A" for the EarthCARE BBR spectral responses shown in Figure 1?

L97, L100, and elsewhere: The paper mentions both "unfiltering factors" and "unfiltering coefficients". I think they are referring to the same quantity. Please choose one term and stick to it to avoid confusion.

L101: Does 5,544 correspond to the number of unique scenes, or does this number include multiple simulations of the same scene at different solar zenith angles? Please clarify in the paper.

L101: Why are there many more simulations for thermal? Since the solar simulations require further stratification (by SZA and RAA) I would expect that having relatively more solar simulations would be beneficial.

L103-108: The reader needs some evidence that the simulations cover the full range of conditions that could be encountered in reality. For example, are the authors confident that the simulations span all combinations of clouds (optical depth, phase, altitude, effective radius, organization, etc), aerosols (optical depth, composition, size distribution, hygroscopicity, etc.), trace gases (tropical, mid-latitude, and polar atmospheres, etc), and surfaces (spectral variability, BRDF, etc)? Very limited information is given. It is not even mentioned where the atmospheric profiles are coming from for the radiative transfer. Full details are needed.

Figure 2: I usually like flow charts to vizualise the products but in this case I am left slightly confused. I see that the B-NOM and B-SNG are provided on different grids/domains, but it is not clear to me why two different product flows are needed. If the B-SNG provides measurements at the detector level, then why not just aggregate the B-SNG radiances over the small/standard/full domains? Also, I do not understand why $L_{LW}$ is used in the B-NOM flow chart, but $L_{TW}$ is used in the B-SNG flow chart. Since $L_{LW}$ is not directly measured, a synthetic $L_{LW}$ is presumably also used with the B-NOM processing. In that case, the two flows are identical other than the final step that deals with the spatial domains, so I must be missing something. To rectify these misunderstandings, I suggest the descriptions of the B-NOM and B-SNG products are further expanded and contrasted in Section 4.1.

Figure 3: A couple of suggestions for improvement:

- It would be clearer if each subplot was labelled individually and referred to in the caption.
- The colours of the data points in the bottom two plots are all red. It would be better to keep the same colour coding as the plots above so that it is easy for the reader to see that the points with the large residuals are the ocean sun glint points.
- The title of the upper right plot says "VZA=50". In the text, it says VZA was 55°, which I expect is correct given the BBR VZA. Please fix this error.

L165-167: As well as the spread from water vapour, it seems that the cloudy points in Figure 4 are generally more to the left of the fit line, whereas the clear-sky points are generally more to the right. If Equation 6 was calculated separately for cloudy and clear (and also possibly separately for tropical, mid-latitude, and polar regions), would that help to reduce the RMS error?

Section 4.5: Since there are no results presented in this section, it doesn't seem to fit. I suggest removing Section 4.5 and mentioning the MSI unfiltering in Section 6 when results are shown, or even just in the conclusions.

Figure 6: It looks like the polynomial fit is not doing very well at capturing the upper end of the thermal radiance values (clear sky). The unfiltering factor shows little dependence on the radiance magnitude beyond 90 W/m2/sr, but the fit shows a sharp increase. Does this create larger errors for the clear scenes? Would it make sense to have a separate LW unfiltering factor fit for cloudy and clear-sky?

L208-210: The relative error values stated in the text do not seem to match those in the table. For SW, the text says ≈0.5% for clear sky but all of the SW clear-sky values in the table are less than this (0.35,0.36,0.42,0.46). For SW cloudy, the text says ≈0.4% but all values in the table are identically 0.34%. For LW the text says "well below 0.1% for all of the scene types", but 4 out of the 10 scenes are at or above 0.1% in the table, and the values below this are only just below. Assuming the values in the table are correct, I suggest updating the text to something like: "For the solar radiation, the relative error on the unfiltered radiances is 0.34% for cloudy conditions and increases to 0.35-0.46% for clear sky conditions. For the thermal radiation, the relative error is 0.10 ± 0.02 % for all scene conditions."

L219-220: The claim that the MSI-based unfiltering does not perform better does not seem to be well supported by the statistics shown in Table 2. The majority of the statistics in Table 2 *are* improved with the MSI-based approach. This also contradicts a statement in the conclusions where it is claimed that MSI radiances are useful to further reduce the unfiltering error.

Table 2: I find it difficult to compare the different example scenes, and also compare to the results in Table 1, because the radiances of the scenes themselves are different. Please include the relative error in % in this table, as was done in Table 1. This will help the comparisons greatly, and is particularly important given that the errors stated in the conclusions and abstract are in %.

Data availability: The doi given to the EarthCARE demonstration products does not seem to include the radiative transfer database used for unfiltering. Per AMT policy, I think the libRadtran simulation database (radiative transfer input profiles and output spectra) should be made available since this is essential underlying data required for this study.

A paper describing the updated unfiltering algorithm for the CERES instruments is now in the public domain: https://doi.org/10.5194/egusphere-2023-1670. The authors did not refer to this paper, which is understandable since it has only been available for about 2 months. However, given the relevance to the EarthCARE unfiltering algorithms, I think the authors need to consider this paper in their revision. It includes several important updates compared to the earlier CERES unfiltering algorithm (already cited). For example, they implemented the Cox-Munk BRDF model over ocean, MODIS retrieved BRDFs over land, considered seasonal variations, increased angular resolution, and used MODTRAN version 5 that has several advantages (see paper for details) to build their simulation database. I recommend adding a paragraph or two comparing and contrasting the EarthCARE BBR approach with this new CERES method. Future users of the EarthCARE BBR data will likely find such a comparison very useful.

***Technical corrections***

L42: "data bases" -> "databases"

L45: remove "the"

---

## Author Comment (AC1)

**Review of "Unfiltering of the EarthCARE Broadband Radiometer (BBR) observations: the BM-RAD product" by Velázquez Blázquez et al.**

**27 September 2023**

**General comments**

The manuscript prepared by Velázquez Blázquez et al. presents an approach to derive unfiltered shortwave (SW) and longwave (LW) radiances from the EarthCARE Broadband Radiometer (BBR). This is an essential processing step to remove unwanted features in the directly measured radiances that are associated with the instrument spectral response. The approach follows that of existing CERES and GERB broadband radiometer measurements that rely on spectral radiance databases simulated by a radiative transfer model. The errors associated with the unfiltering process are reported to be below 0.5% and 0.1% in the SW and LW, respectively.

The paper will serve as a useful reference within the scientific literature for any future users of the EarthCARE BBR data. Overall the paper is clear and logical, well written, and supported with appropriate figures. I noticed some inconsistencies with the EarthCARE overview paper that is also part of this special issue. There are several other aspects that I think would benefit from further clarifications and explanations, especially regarding the radiative transfer simulations. There is also a new paper for CERES unfiltering that should be considered. After addressing these concerns, as outlined in my comments below, I recommend publication in AMT.

**Specific comments**

L20: In Wehr et al. 2023, the spectral limits of the BBR SW and TW channels are stated as < 0.25 to 4.0 μm for SW and 0.25 to > 50 μm for TW. Here, the spectral limits are stated as 0.25 to 5 μm for SW and 0.25 to 500 μm for TW. Since both studies are new, I expect the limits have not changed, rather there is an error somewhere. In most other studies, this might seem like a picky comment because there is probably very little energy difference between the two sets of spectral limits. However, since the purpose of this study is spectral unfiltering, the instrument spectral limits seem like a basic characteristic to ensure is correct and consistent.

The spectral limits have not changed and they are indeed well defined by the spectral response of the instrument as shown in Figure 1. As shown, there is not a sharp limit, but the authors agree with the comment from the reviewer and will update the limits to those defined by Wehr et al. 2023 for consistency between the two papers.

L21: Similar to the comment above, the stated spatial resolution of the detector array (700m along track and 600 m across track) is inconsistent with that stated in Wehr et al. 2023 (648 m along track, 800 m across track). Please check that stated numbers are correct and consistent. Do these numbers represent resolution, or sampling distance? It also should be noted that these numbers are relevant to nadir only.

Thanks for pointing this out, the numbers have been verified and updated the detector array size to 648m along and across track. Note that 800 m in the paper of Wehr et al refers to the sampling distance.

L31: I calculate that, at an EarthCARE altitude of 393 km with an orbital period of 92.5 min, the duration between fore and aft 55° views of the surface is 2.79 minutes. If the authors agree, it would be better to update "about 2 minutes" to "about 3 minutes".

Thank you for pointing this out. We had written about 2 minutes because according to the technical documentation of the BBR, EC-AN-SEA-BBR-0020 Integrated energy analysis document: "The temporal separation between a SW and TW capture of a scene with the same telescope is about 60ms; the temporal separation between a nadir and an oblique capture of the same target is about 70s", therefore 140s (2.33 min) but we agree with your comment and calculation and the text have been updated to the 3 min proposed.

L32: Similar to the first two comments above, the stated swath of the detector array ($\sim$ 17km for the nadir view and $\sim$28 km for the two oblique views) is inconsistent with that stated in Wehr et al. 2023 (±10.2 for nadir and ±16 for off-nadir views). Please ensure values are correct and consistent. This is information that data users are likely to pick up on and, once published, incorrect numbers can be easily propagated into other works.

We confirm the swath of the detector array is fine in our article.

Each telescope uses an array of 30 microbolometer detectors, allowing an across-track swath of $\sim$ 17 km for the nadir view and $\sim$28 km for the two oblique views.

L28: There is a reference to another paper that describes the BMA-FLX processor (Velázquez-Blázquez et al., 2023), but as far as I can tell this paper is not available anywhere online. It would have been good for the reviewers to at least see a draft copy of this paper if it is to be cited here.

The BMA-FLX paper is intended to be part of the EarthCARE Special Issue, and at the time of writing it was not finalized. The output of the BM-RAD processor, described in this paper is meant to be used as the main input in the BMA-FLX processor, this is why this paper is cited here.

L59-60: For the CERES instrument flying on the NOAA-20 satellite, there is a dedicated LW channel. So, the statement that LW is calculated by subtraction is not always true.

We completely agree, fixed.

Equation 6: What is the value of "A" for the EarthCARE BBR spectral responses shown in Figure 1?

The value is now A=1.08511561332069467257 (for the nadir view) but as this value is subject to change during the mission, following recomputations of the Spectral Responses due to ageing, we prefer not publish a fix value that can become inaccurate in the future.

L97, L100, and elsewhere: The paper mentions both "unfiltering factors" and "unfiltering coefficients". I think they are referring to the same quantity. Please choose one term and stick to it to avoid confusion.

Agree and corrected in the text.

L101: Does 5,544 correspond to the number of unique scenes, or does this number include multiple simulations of the same scene at different solar zenith angles? Please clarify in the paper.

616 unique scenes and including solar geometry then 5544. Clarified in the text.

L101: Why are there many more simulations for thermal? Since the solar simulations require further stratification (by SZA and RAA) I would expect that having relatively more solar simulations would be beneficial.

You are right, but this is motivated by the fact that for the LW simulations there is a higher variability in atmospheric profiles and in the surface temperatures used in the simulations.

L103-108: The reader needs some evidence that the simulations cover the full range of conditions that could be encountered in reality. For example, are the authors confident that the simulations span all combinations of clouds (optical depth, phase, altitude, effective radius, organization, etc), aerosols (optical depth, composition, size distribution, hygroscopicity, etc.), trace gases (tropical, mid-latitude, and polar atmospheres, etc), and surfaces (spectral variability, BRDF, etc)? Very limited information is given. It is not even mentioned where the atmospheric profiles are coming from for the radiative transfer. Full details are needed.

Indeed, having a database that realistically represent the conditions to be observed by BBR is crucial for this study. The authors consider that the simulations covered a significantly wide range of surface and atmospheric conditions. This is not included in the manuscript because the justification and details of the RT simulations are provided in a published technical note (i.e., Velazquez et al. 2010). Please note that the reference is already cited in the text, and link to the document is now included in data availability and in the references.

Figure 2: I usually like flow charts to vizualise the products but in this case I am left slightly confused. I see that the B-NOM and B-SNG are provided on different grids/domains, but it is not clear to me why two different product flows are needed. If the B-SNG provides measurements at the detector level, then why not just aggregate the B-SNG radiances over the small/standard/full domains? Also, I do not understand why LLW is used in the B-NOM flow chart, but LTW is used in the B-SNG flow chart. Since LLW is not directly measured, a synthetic LLW is presumably also used with the B-NOM processing. In that case, the two flows are identical other than the final step that deals with the spatial domains, so I must be missing something. To rectify these misunderstandings, I suggest the descriptions of the BNOM and B-SNG products are further expanded and contrasted in Section 4.1.

It is a very pertinent comment but you must know that the authors are not involved in the development of the L1 BBR products performed by the industry. The B-NOM and B-SNG products are both in the same grid, the BBR grid, however, in B-NOM they provide SW and LW radiances and in B-SNG they provide SW and TW. As a L2 developers we have chosen to use both B-NOM, with the defined domains, i.e., standard, full and small and, in addition, develop a configurable assessment domain in the JSG from the single pixel BBR measurements from B-SNG for the closure assessment.

Figure 3: A couple of suggestions for improvement:

• It would be clearer if each subplot was labelled individually and referred to in the caption.

• The colours of the data points in the bottom two plots are all red. It would be better to keep the same colour coding as the plots above so that it is easy for the reader to see that the points with the large residuals are the ocean sun glint points.

Good suggestion. Updated accordingly.

• The title of the upper right plot says "VZA=50". In the text, it says VZA was 55°, which I expect is correct given the BBR VZA. Please fix this error.

Fixed.

L165-167: As well as the spread from water vapour, it seems that the cloudy points in Figure 4 are generally more to the left of the fit line, whereas the clear-sky points are generally more to the right. If Equation 6 was calculated separately for cloudy and clear (and also possibly separately for tropical, mid-latitude, and polar regions), would that help to reduce the RMS error?

Yes, indeed that could probably help to reduce the RMS error but will introduce complexity due to the needed MSI cloud information. This could be tested using night time data for which the SW channel provides the contamination due to the absence of solar radiation.

Section 4.5: Since there are no results presented in this section, it doesn't seem to fit. I suggest removing Section 4.5 and mentioning the MSI unfiltering in Section 6 when results are shown, or even just in the conclusions.

The results of the MSI-based approach are presented in the Table 2. This is now mentioned in this section.

Figure 6: It looks like the polynomial fit is not doing very well at capturing the upper end of the thermal radiance values (clear sky). The unfiltering factor shows little dependence on the radiance magnitude beyond 90 W/m2/sr, but the fit shows a sharp increase. Does this create larger errors for the clear scenes?

Indeed, the fit doesn't perform well for very high radiances, but the error in the LW unfiltering remains lower than 0.3% in the worst case. This will be monitored with real data and if needed a more complex fit will be adopted for very warm scenes.

Would it make sense to have a separate LW unfiltering factor fit for cloudy and clear-sky?

Given the good results in the LW unfiltering it doesn't seem needed to introduce a dependency in the cloud products.

L208-210: The relative error values stated in the text do not seem to match those in the table. For SW, the text says ≈0.5% for clear sky but all of the SW clear-sky values in the table are less than this (0.35,0.36,0.42,0.46). For SW cloudy, the text says ≈0.4% but all values in the table are identically 0.34%. For LW the text says "well below 0.1% for all of the scene types", but 4 out of the 10 scenes are at or above 0.1% in the table, and the values below this are only just below. Assuming the values in the table are correct, I suggest updating the text to something like: "For the solar radiation, the relative error on the unfiltered radiances is 0.34% for cloudy conditions and increases to 0.35-0.46% for clear sky conditions. For the thermal radiation, the relative error is 0.10 ± 0.02 % for all scene conditions."

Yes indeed, thanks for your comment. The text was modified as suggested following values updated in the table.

L219-220: The claim that the MSI-based unfiltering does not perform better does not seem to be well supported by the statistics shown in Table 2. The majority of the statistics in Table 2 are improved with

the MSI-based approach. This also contradicts a statement in the conclusions where it is claimed that MSI radiances are useful to further reduce the unfiltering error.

*Indeed, the MSI-based unfiltering performs better, but not in a significant way, this is why it is written that it does not perform significantly better. We do not see the contradiction with the conclusions.*

Table 2: I find it difficult to compare the different example scenes, and also compare to the results in Table 1, because the radiances of the scenes themselves are different. Please include the relative error in % in this table, as was done in Table 1. This will help the comparisons greatly, and is particularly important given that the errors stated in the conclusions and abstract are in %.

*We agree, however, some specificities of the simulated data prevent us to make a full quantitative comparison between the scenes. For instance, solar radiances over ocean are too dark with simulated radiances as low as 15 W m -2 sr-1 in clear conditions for the nadir view. Also, the simulated radiances are limited in terms of wavelength range (0.2-4µm for the SW and 4-400 µm), which introduce some artificial error in the estimation of the inter-channel contamination. Furthermore, Halifax and Baja scene have a systematically high solar zenith angle which makes the relative error important on those simulations (~1% for the SW). Still we think it is important to provide the Table 2 as the RMS errors show that the unfiltering performs well within the mission requirements or 2.5 Wm-2sr-1 for SW and 1.5 Wm-2sr-1 for LW.*

Data availability: The doi given to the EarthCARE demonstration products does not seem to include the radiative transfer database used for unfiltering. Per AMT policy, I think the libRadtran simulation database (radiative transfer input profiles and output spectra) should be made available since this is essential underlying data required for this study.

*The link to the radiative transfer database and description has been added to the data availability.*

A paper describing the updated unfiltering algorithm for the CERES instruments is now in the public domain: https://doi.org/10.5194/egusphere-2023-1670. The authors did not refer to this paper, which is understandable since it has only been available for about 2 months. However, given the relevance to the EarthCARE unfiltering algorithms, I think the authors need to consider this paper in their revision. It includes several important updates compared to the earlier CERES unfiltering algorithm (already cited). For example, they implemented the Cox-Munk BRDF model over ocean, MODIS retrieved BRDFs over land, considered seasonal variations, increased angular resolution, and used MODTRAN version 5 that has several advantages (see paper for details) to build their simulation database. I recommend adding a paragraph or two comparing and contrasting the EarthCARE BBR approach with this new CERES method. Future users of the EarthCARE BBR data will likely find such a comparison very useful.

*We have had a close look at this interesting paper that provides several improvements with respect to the current CERES unfiltering process and we are looking forward to the implementation of this work in the CERES processing. We propose to add the following sentence: It is worth to mention that a CERES team is currently reviewing its unfiltering process and a several improvements are proposed in Liang et al. (2023) for possible inclusion in Edition 5.*

Technical corrections L42: "data bases" -> "databases"

*Corrected*

L45: remove "the"

Removed

---

## Author Comment (AC2)

**Answers to review https://doi.org/10.5194/amt-2023-170-RC2**

**General points**

The manuscript describes the algorithm to obtain spectrally unfiltered shortwave (SW) and longwave (LW) radiances at top-of-atmosphere from the observations by the BBR instrument on board of EarthCARE. The spectral unfiltering is needed to correct for the spectral sensitivities of the instrument and for the contamination of SW radiation in the LW channel and vice versa. This paper is part of the special issue describing the algorithms for EarthCARE.

The structure and overall story of the manuscript is quite clear. However, when it comes to the details of the algorithm, the equations, and the steps that are taken, the paper is not clear and sometimes ambiguous, especially in Section 2. Since the BM-RAD algorithm should be perfectly clear to avoid errors later on, corrections in the manuscript are needed. These changes are not very difficult but necessary, and therefore these are major revisions. Furthermore, several technical corrections should be applied throughout the paper.

**Specific points**

- In the Introduction (Section 1) more high-level information on the aim of the BBR instrument in combination with other EarthCARE instruments is needed.

Thanks for your comment. Indeed, a brief introduction about the BBR role in the EarthCARE mission would help the reader understanding better the importance of the paper. Below the new paragraph included:

*The EarthCARE (Earth Clouds, Aerosols and Radiation Explorer) mission (Illingworth et al., 2015; Wehr et al., 2023) is a collaborative mission between the European Space Agency (ESA) and the Japan Aerospace Exploration Agency (JAXA). EarthCARE's primary objective is to enhance our understanding of the processes affecting clouds, aerosols, and radiation in Earth's atmosphere. The mission aims to provide valuable information for improving climate model parametrizations and the understanding of how these components influence the global climate. EarthCARE integrates a suite of instruments including a lidar, radar, and radiometric instruments. Among these instruments, the Broadband Radiometer (BBR) plays the role of providing crucial information for the radiative closure of the mission. This process involves verifying that the radiative transfer simulations, which are fed with atmospheric products from the mission's active sensors, report radiative fluxes within 10 W m2 of the fluxes derived from the BBR-*

- The paragraph starting on L. 29 seems too detailed for the Introduction.

Thank for your suggestion. However, we consider relevant to include this information in the paper as it briefly describes the instrument and the EarthCARE product where the radiances are stored.

- L. 25: what is the function phi? Please mention that it is the spectral response function, and is needed to separate between reflected solar and emitted thermal fluxes.

Already clarified in the text.

- L. 29: all three references should be between brackets.

Thank you, corrected.

- Section 2: This section contains bugs, is unclear and should be improved and clarified. The symbols are unclear, the different unfiltering steps are not clearly separated.

We have modified the section following the comments from the reviewer, clarifying symbols and correcting bugs.

- Equation 1: please give the integration limits; this holds for all integrations.

Fixed.

- L. 58: SW channel: do you mean SW radiation ?

It is not the SW radiance, but the SW channel. It is difficult to manufacture a LW filter as opposed to the quartz filter used for the SW channel.

- L. 58: efficient > an efficient

Corrected.

- L. 62, Eq. 5: subscript SW - in other places you use sw. Please be consistent in the subscripts throughout the paper. SW and LW are clearer subscripts than sw and lw.

Thanks for spotting the typos. Typo in subscript corrected in equation 5. Following the recommendation of the reviewer the subscript sw has been replaced by SW, lw by LW and tot by TW.

- L. 64: observed. > observed:

Corrected.

- Eqs. 7 and 8: I find it confusing that the spectrally integrated radiance has the same symbol L as is used for the spectrally dependent $L(\lambda)$. Please use a clear distinction in symbols.

We appreciate this comment from the reviewer, but please note that the notation is used through several official documents and, therefore, we would prefer not to introduce a new symbol to make the difference between the spectral and integrated radiances. Every time any spectral quantity is mentioned it is followed by (lambda).

- L. 76-79: "This conversion .... solar radiation": Please clearly separate these 3 factors.

Rephrased to consider the referee comment.

- Eqs. 8-9: The terminology in these equations, e.g. the subscripts "sw,sol" and "lw,th", is not clear.

We understand it refers to Eqs 9-10. Clarification is included in the text for the subscripts.

- L. 85: L_sol,L_th: where is now the subscript unfil, which was used in Equations 7-10 ?

Subscript unfil has been removed for clarity in equations 7-10

- L. 93-94: please clearly separate these different steps.

Rewritten.

- Eqs. 13-14: what happened to the fil and unfil subscripts introduced in earlier equations?

Removed also from previous equations.

- L. 97: alpha_sw, alpha_lw: these are new variables! Where are alpha_sol and alpha_th introduced in Eqs. 9-10?

Typo in Eq 9 and 10 corrected. Now notation is consistent everywhere using alpha_SW and alpha_LW

- L. 97: the name "unfiltering process" is quite confusing: there is an unfiltering step and a decontamination step. Please clarify the entire Section 2.

We agree that the terminology can be misleading but it is commonly used in the field. The unfiltering process includes the contamination removal and the unfiltering itself.

- L. 102: double bracket )

Corrected.

- L. 103: physical > geophysical

Corrected.

- L. 104: remove , in such as,

In our opinion, this is not needed, as it clarifies that the ancillary models and data are for instance the surface reflectances from the Aster Spectral Library data (Baldridge et al., 2009) and the Optical Properties of Aerosols and Clouds (OPAC) software

- L. 106: aerosols: how about clouds? what are the ranges in optical thickness and height of aerosols and clouds?

Some more details have been added to the manuscript, but full description of the radiative transfer simulation database is available at
https://gerb.oma.be/public/almudena/SITS_DB_compressed/GeoType_data_base_desc.pdf

- L. 114: droplets

Typo corrected.

- L. 119: are the three spectral response functions of the instrument independent of the (3) viewing directions?

Yes, each view has its own spectral response

- L. 136: MSI-based algorithm

Updated

- Figure 2: please use the same font style for the symbols and equations in this figure.

Fixed

- L. 140: The Fig. > Figure; same comment on L. 156.

Corrected

- L. 148: It is quite remarkable that alpha does not depend on cloudiness. What is the explanation that the solar contamination in the longwave channel does not depend on cloudiness, since thick clouds in the daytime reflect a lot of radiation.

Indeed, the solar contamination depends on the cloudiness, but the alpha coefficient appears to be independent of the cloudiness type.

- L. 150: m/s; note that all units should be written in upright font.

Corrected

- Fig. 3: the red colored dots in the two lower plots are unrelated, I assume, to the legend of the upper plots. Then please make these dots black, and define the residual in the main text.

Lower plots updated to match the legend of the upper ones

- Eq. 16: symbol a is already used in Eq. 15, and has a different meaning there. Please use unique symbols for each quantity. The same remark holds for the next equations. Please be consistent in symbols and terminology.

Fixed.

- Fig. 4: On the basis which points did you determine the precise relationship shown in the 4 plots? The a and b coefficients are exactly the same, but the data points are different. That seems strange.

Caption adapted to make clear that the same fit is used for the 4 surface types.

- Caption Fig. 4: please explain the two types of points in each subplot.

Done in the caption.

- Sect. 4.4, first sentence: Please explain where we are in the procedure of the flow diagram. Do you mean spectral unfiltering to obtain the correct SW radiances? Is this the step after the decontamination?

Clarified in the document.

- Eq. 17: apart from the reuse of earlier symbols, this is an unclear formula. Please make a multiplication with the inverse or add brackets.

Formula rewritten.

- Fig. 5: the plots and their legends are poorly readable. Please use a larger font.

Fixed

- L. 190: stand-alone algorithm

Added.

- L. 196: "much smaller than for the CERES and GERB instruments": what is the reason?

Reason added: The primary reason for this is that the BBR optics has only one mirror while CERES has two and GERB has five.

- Fig. 6: what are the fit coefficients? This fit function does not hold for larger radiances.

Indeed, the second order polynomial fit doesn't fit very well the scenes with very high radiances, in which an error of ~0.3% could be introduced in the worst case. That was considered acceptable given the low occurrence of such a high thermal radiance.

- Caption Fig. 6: use correct ordering: (a) ..., (b) .....

Corrected

- Caption Fig. 6: please explain: are these the alpha factors of Eq. 10 or of Eq. 14?

Equation referenced in the caption.

- Table 1 header: what is the reason that you switch between LW and th, and between SW and sol?

Changed SW to solar and LW to thermal

- Section 5: Section 5 is very short. How did you do the analysis? Please explain how you arrive at Table 1.

Yes, indeed the section is relatively short, but it must be understood as a high level verification that the algorithm performs as expected. The full verification of the algorithm is done using test scenes in section 6.

- Title Section 5: do you mean algorithm verification instead of performance verification? The next section is about performance verification.

Yes, indeed, thanks for your comment. Title updated to BM-RAD algorithm verification.

- L. 217: Table 2

corrected

- L. 225: Please summarize the results of Fig. 7: which conclusions do you draw from it?

Some more information has been added in the document. A summary of the conclusions can be found in section 7.

- Fig. 7: The plots in the right column are somewhat smaller than in the other columns. Please make all subplots the same size. How will you orient this figure to make it readable? Preferably landscape.

Figure now in landscape mode.

- Caption Fig. 7: "RT sim (truth)"

Corrected.

- References: Please correct all references, because the initials should be put after the surname.

Corrected.

Technical corrections throughout the manuscript

- level 1 > level-1, level 2 > level-2

All corrected.

- Subscripts which are words, abbreviations or acronyms should be in upright font. For example, fil in $L\_fil$ on l. 50 should be upright. This occurs many times in the manuscript.

Fixed

- Symbols should be in italic font.

Corrected

- L. 67: Fig. 5 > Figure 5. If it is the starting word of a sentence, Fig. should be written in full.

Corrected

- Always use a space between number and unit: e.g. L. 91: 5 μm

Corrected where needed.

- Units should be written in upright font (e.g. l. 152, and many other places)

Fixed

- Fig., Eq., Sect. should be written with capital.

Verified and corrected where needed.

- Tables: please put the table caption above the table.

Done

Citation: https://doi.org/10.5194/amt-2023-170-RC2

---

## Author Response (AR2)

**Answers to review https://doi.org/10.5194/amt-2023-170-EC1**

Dear authors and reviewers,

Thank you to the reviewers for their first set of comments and to the authors for their revisions. I am going to send the manuscript out for a second review, and the reviewers/authors should take note of my comments below in the second reviews/revisions.

We thank the editor and reviewers for their comments. Below are the original comments in black with our responses in blue.

**REVIEWER 1**

1. Regarding the comment on the flowcharts in Fig. 2, readers of this paper are very likely to have the same questions as the reviewer, so the authors should update the text to add clarity and explanations. The reviewer is not asking why the B-NOM and B-SNG products are the way they are (indeed their design is out of your hands) but to explain the reasoning for BM-RAD to contain data from two different sources (B-NOM and B-SNG), and explain how you envision the two being used - perhaps different applications will require different sources?

Sentence added to clarify why it is important the Assessment Domain resolution for the radiative closure evaluation within EarthCARE: *"Among the different resolutions, the AD is especially important as the EarthCARE radiative computations products (Cole et al., 2023) will be evaluated on this domain."*

Reference added to the manuscript.

2. Further to my comment 1, the caption of Fig. 2 could be clearer, because it implies the flowchart is for the processing carried out in the production of B-NOM and B-SNG, when in fact it is the processing applied to B-NOM and B-SNG data to produce BM-RAD (which contains both outputs?)

Agreed and modified as suggested. Caption is now: *"Unfiltering flowchart for the BM-RAD product on the BBR grid resolutions (Full, Standard and Small) from the level-1 B-NOM (left panel) and the on the JSG grid resolutions (Assessment, JSG and JSG enhanced) from the level-1 B-SNG product (right panel)"*

3. Figure 3: please label all four panels as (a) to (d), and refer to them by letter in the caption.

Modified as suggested.

4. Regarding the comment on L219-220 of the original manuscript, I agree with the reviewer that the text is inconsistent with both the table and the conclusions. The new text in section 6 reads:

"The error metrics show that the MSI-based shortwave unfiltering does not perform significantly better than the stand-alone unfiltering approach. The gaining of including MSI information in the unfiltering process while improving results is not very relevant."

The conclusions state:

"Scene information from the MSI radiances (from M-RGR product), MSI cloud retrievals (from M-CLD processor), or snow products (from X-MET product) are useful to further reduce the unfiltering error."

These are inconsistent. Why not state in section 6 that a small improvement is detected but that the difference might not be significant in practice because of parallax effects (which you state in the conclusions)? The current use of the word "significant" in section 6 implies you are talking about formal statistical significance, but I don't think you have tested this. The word "relevant" also doesn't seem correct - surely any improvement of accuracy is "relevant"? Perhaps you mean "large" (as in "not very large")?

*The text has been clarified following the recommendation of the editor. The text is now:* "The error metrics show that the MSI-based shortwave unfiltering provides in general a small improvement. The gaining of including MSI information in the unfiltering process while improving results might not be very large in practice because of parallax effects."

**REVIEWER 2**

5. Regarding the definition of \phi(\lambda), if you use it in the introduction, it must be defined in the introduction not in the section after. I suggest changing the text simply to "measured by a perfect instrument, i.e. one with a flat spectral response $\phi(\lambda)=1$ (where $\lambda$ is wavelength), ..."

Agreed and updated.

6. Regarding the comment on the original L148, I think the explanation that alpha does not depend on cloudiness is that clouds are not reflective in the longwave, which could be stated in the manuscript. Indeed, if you plug cloud single-scattering albedo and asymmetry factor into the equation for the reflectance of a cloud in the limit of infinite optical depth (e.g. Eq. 13.45 of Petty's book on atmospheric radiation) then you get a value typically in the range 0.02-0.1 in the thermal infrared. So there is much less contrast with the underlying surface than you get in the shortwave.

As pointed out by the editor, clouds are not reflective in the longwave. However, the solar contamination in the LW channel as discussed in this section is the result of the definition of the synthetic LW channel which has negative "sensitivity" in the SW region (in such a way that the LW for a solar spectrum of 5800 K is zero). This contamination, is therefore negative and proportional to the solar radiances and relatively independent of the spectral signature of the scene. This is the reason why the alpha factor does not depend on the cloudiness.

**OTHER COMMENTS**

7. Figs. 4 and 5: Yellow is a poor choice of colour as it appears quite faint - can you use a darker shade, e.g. orange?

Figs. 4 and 5. have been updated using a darker shade of yellow, as orange was already used for another surface type. Blue and green colors have also been modified to improve the plot.

8. L248 of revised manuscript: the reference to Fig. 7 appears as "Fig. ??"

Missing reference to the figure added.